# Current Aspects Regarding the Clinical Relevance of Electroacupuncture in Dogs with Spinal Cord Injury: A Literature Review

**DOI:** 10.3390/ani11010219

**Published:** 2021-01-18

**Authors:** Madalina Florina Dragomir, Cosmin Petru Pestean, Iulia Melega, Cecilia Gabriella Danciu, Robert Cristian Purdoiu, Liviu Oana

**Affiliations:** 1Department of Surgical Techniques and Propaedeutics, University of Agricultural Science and Veterinary Medicine Cluj-Napoca, Calea Manastur no. 3-5, 400372 Cluj, Romania; cosmin.pestean@usamvcluj.ro (C.P.P.); iulia.melega@usamvcluj.ro (I.M.); oanaliviu2008@yahoo.com (L.O.); 2Department of Physiology, University of Agricultural Science and Veterinary Medicine Cluj-Napoca, Calea Manastur no. 3-5, 400372 Cluj, Romania; cecilia.danciu@usamvcluj.ro; 3Department of Clinical Sciences, University of Agricultural Science and Veterinary Medicine Cluj-Napoca, Calea Manastur no. 3-5, 400372 Cluj, Romania

**Keywords:** acupuncture, complementary therapy, canine, disc disease

## Abstract

**Simple Summary:**

This review aims to provide an overview of the relevance of electroacupuncture in dogs with spinal cord injury. We illustrate and discuss the areas that have advanced in recent years and those that still require further research. Systematic studies have been conducted on Google Scholar, PubMed, Science Direct, Cab Direct, and Research Gate. This article is believed to be the first review of electroacupuncture in dogs with spinal cord injury. Electroacupuncture provides better analgesia than direct acupuncture by increasing spinal opioid release and increases blood supply to the spinal cord and nerve root. When combined together, Eastern and Western treatment offers faster recovery and improved ambulation and perception of deep pain than medical treatment results alone.

**Abstract:**

In recent years, the use of acupuncture and electroacupuncture has been increasing as more clinical research has been conducted showing positive results in the treatment of animals, particularly dogs. Electroacupuncture is a more complex and specific acupuncture technique that involves electrical stimulation on acupuncture needles. Most of the studies have shown that the beneficial effects of electroacupuncture are more evident than in acupuncture alone. This review included studies focused only on dogs with spinal cord injury. Research facility animals (mice, rats, and rabbits) were avoided. Titles and abstracts of identified articles were read, and outlines were made to be better understood. Clinical applications are discussed and suggested in each section. When specialists use this method, electroacupuncture can be an excellent complementary therapy for veterinary patients’ pain control.

## 1. Introduction

Electroacupuncture (EA) is a specific acupuncture technique that involves electrical stimulation applied to acupuncture needles, and it has been recommended to treat various painful conditions, including neurologic and muscle deficits [1].

Acupuncture (Ac) has the role of the stimulation of well-defined points on the skin by using needles. A relevant approach like EA, laser acupuncture, or acupressure is often used under the umbrella term [2]. The basic principle of these medical advances is that disorders related to the “Chi” (Qi: vital force or energy) movement can be managed by stimulating various points on the body’s surface [3].

Intervertebral disc disease (IVDD) has begun to become more common in pets. The most common reported symptoms depending on the degree of neurological dysfunction are graded from 1 to 5, where grade 1 is assigned for dogs with no neurologic signs, only pain correlated with IVDD, and grade 5 is assigned for dogs with no deep pain perception and paraplegia [4].

Today, learning Traditional Chinese veterinary medicine (TCVM) requires a shift in perspective. In general, Western veterinary medicine (WVM) represents all the procedures we use to diagnose and treat symptoms and diseases, like drugs, imaging, or surgery. TCVM believes more in balancing the whole-body Qi energy and the body’s ability to heal itself. They are merely two different ways of viewing the world, and each system has its strengths and weaknesses. Both hope to promote health and prevent disease [5].

The present paper aims to review the relevant literature on EA’s clinical relevance in dogs with spinal cord injury.

## 2. Current Methods

### 2.1. Research Data

Five essential subject resources were used: Google Scholar, PubMed, Science Direct, Research Gate, and CAB-Direct. Literature research was described to collect all articles with the following keywords: Ac, complementary therapy, disc disease, and veterinary medicine. Thus, 3270 studies have been identified, out of which 2020 were related to spinal cord injuries. As we refined our search to intervertebral disc disease (IVDD), the results further narrowed to 104 articles. Studies using laboratory animals, as well as studies that included aqua-acupuncture and equal publications, were excluded. After excluding all irrelevant articles, only nine articles that meet our criteria remained.

### 2.2. Inclusion and Exclusion Criteria

Only articles about dogs diagnosed with different stages of IVDD and whose treatment contained EA were included in this review. Dogs had to have complete clinical data history available and final diagnosis provided by imaging techniques, such as magnetic resonance imaging or computed tomography scans. Some patients underwent spinal decompression surgery (hemilaminectomy) to compare the degree of ambulation offered by EA postoperatively to conventional corticosteroid drugs. No surgery was performed for the rest of the patients, but the degree of neurological recovery was studied using EA compared to corticotherapy. Furthermore, the short-term and long-term outcomes of each study’s different therapy options had to be available for review. Studies other than those conducted on dogs were excluded from this review.

In our review, we separated the patients into five major groups as follows: dogs that went under surgery represented Group A, Group B consisted of dogs that underwent surgery followed by EA, with Group C consisting of dogs treated with corticosteroids only. Dogs that underwent corticotherapy and EA represented Group D. Group E was the group that had only EA.

### 2.3. Data Abstraction

The data collected from articles were divided according to the time of treatment and the recovery rate.

## 3. Results

Nine studies met the inclusion criteria. Two hundred twenty-six dogs were included in all these studies. Each study group had a well-described treatment protocol. All studies had different clinical stages in treating IVDD in dogs; only six out of nine studies described the degree of neurologic dysfunction (grade 1 to 5). Therefore, four studies included only dogs with a loss of deep nociception. Two studies included only dogs with paresis and paraplegia, and four out of nine studies described treatment options after surgical management of IVDD. All but ten of the patients received corticosteroids. Each study has a well-described treatment in Table 1, except for the 4th study, which is debated in the Discussion section.

The first study was conducted on 20 dogs with ambulatory paresis due to induced spinal cord injury with grade 2 and 3 of neurological dysfunction after 25% of the spinal cord was compressed. The dogs were divided into Groups C, D, and E; the control group had no treatment. All dogs received corticosteroid treatment, except for Group E and the control group. There were no significant differences between Group C and Group E. However, Group D’s recovery days were significantly shorter than the other groups’. According to these results, it was considered that Group D was therapeutically more effective for dogs with ambulatory paresis diagnosed with spinal cord injury [4].

The second study was conducted on 50 dogs with different neurological disorders from grades 1 to 5. All dogs were assessed for functional neurology score (F.N.S.) once a week for at least three weeks, and the last assessment was done by the same researcher who performed the EA. In this study, the dogs were divided into Groups C and D. A numerical test score for the neurological function was used to evaluate the results. Success was considered when a dog with high-grade disorders could walk without difficulty and had a good improvement in conscious proprioception. The treatment was more effective for Group D and led to outpatient recovery and in-depth pain perception in dogs with signs of thoracolumbar IVDD in a shorter time than in the case of Group C [6].

The third study was conducted on a non-ambulatory dog with grade 3 IVDD and tetraparesis and cervical pain, diagnosed with multiple bulging discs, and who had received one dose of prednisone which was then replaced with meloxicam. It was involved in Group D. The dog experienced recovery of proprioception and locomotion with only a slight deficiency in one of the thoracic limbs after ten treatment sessions with EA and Chinese herbal medicine. The patient received one session of EA, once a week for eight weeks. The last two sessions were made every two weeks; in total, there were ten sessions in 12 weeks [7].

The fourth study aimed to compare the severity of postoperative pain in 15 dogs undergoing hemilaminectomy due to acute grade 3 and 4 IVDD. The dogs were divided into Groups A and B. All dogs received pre-surgical corticosteroids and post-surgical fentanyl, but the dose of fentanyl was much lower in Group B than in Group A. The pain score showed small differences between Group A and Group B. The result provided unequivocal evidence that the EA adjuvant could provide some slight benefits. The purpose of this study was to monitor postoperative pain management [8].

The fifth study’s objective was to compare the effects of decompression surgery, EA, and a combination of both for the treatment of thoracolumbar IVDD in 40 dogs with longstanding neurologic deficits (>48 h). The thoracolumbar spinal cord injury was classified based on neurologic signs using a scale ranging from 1 (least severe) to 5 (most severe). Dogs were classified via a modified myelopathy scoring system. Dogs that underwent decompression surgery represented Group A. Group D received EA, and Group B underwent surgery followed by EA. The result was taken into account when the patient initially diagnosed with a high degree of deficiency recovered significantly six months after the treatment. All dogs received corticosteroid treatment. This study showed that Group D was more effective than Group A in recovering ambulation and improved neurologic deficits when performed ≥48 h after the onset of clinical signs [9].

The sixth study was conducted on 80 paraplegic dogs with grade 2 and 3 IVDD and intact deep pain perception. The dogs reviewed for this investigation were divided into Group C and Group D. The results suggest that Group D was more effective than Group C in the recovery of ambulation, back pain relief, and relapse reduction [10].

In the seventh study, three dogs were diagnosed with paralysis of pelvic limbs without deep pain, normal defecation, and urination due to severe traumatic spinal cord injury, and all three underwent spinal cord decompression surgery. All three dogs were assigned to Group B and were diagnosed with grade 3 and 4 IVDD. All dogs regained deep pain perception and presented tail wagging at 14–35 days after being treated with EA and Chinese herbal medicine (The Duhuojisheng-tang consists of 16 herbs including Radix Aralia Contientalis and LoranthiRamulus); some of the dogs recovered almost normal ambulation, urination, and defecation ability. It was considered a success when the dogs recovered from grade 5 to grade 2 of neurological dysfunction. One dog failed to regain normal ambulation due to postoperative inflammation complications [11].

In the eighth study, one dog was evaluated for severe pelvic limb ataxia, atrophy, and paresis and was assigned to Group D. It was diagnosed with multiple thoracolumbar grade 5 IVDD. The dog was treated for one month with corticosteroids with no amelioration seen. After six months of treatment with EA and Chinese herbal medicine (Duhuojisheng-tang), the dog had improved to grade 1 of neurological dysfunction with better mobility, proprioception, and spinal posture [12].

All 16 dogs included in the ninth study were diagnosed with grade 5 IVDD along with the absence of proprioception and failure of hop test, absence of superficial and deep pain perception, and increased spinal reflexes, along with urinary and fecal incontinence at the beginning of the study. Despite mild neurological and functional improvements, no conclusive beneficial effect could be associated with stem cells derived from canine exfoliated deciduous teeth (SCED). There were no major differences between Group A and Group B [13].

In the reviewed articles, 226 dogs were included in the studies. Five patients were in the control group and did not receive any treatment. Five patients received only EA and were included in Group E. A total of 216 patients (95.57%) were treated with corticosteroids and were assigned to Groups A, B, C, or D, depending on the treatment plan. Different dosages were used for all these dogs; 187 were treated with prednisone (dose: 1 mg/kg), and 29 were treated with methylprednisolone (dose: 30 mg/kg).

Three studies (3rd, 7th, and 8th) reported EA and Chinese herbal medicine treatment results in dogs with severe spinal cord injury. They were diagnosed with grade 3, 4, and 5 IVDD and were assigned to Groups B and D. There was a recovery of ambulation without assistance in four out of five dogs. One dog failed because of postoperative complications. All five dogs also received corticotherapy.

The 4th, 5th, 7th, and 9th studies reported treatment including decompressive surgery alone or in combination with EA. Seventy-four dogs were included, from which 25 were in Group A and 30 in Group B. All dogs received corticosteroids. Eight dogs out of 25 were treated with surgery-recovered ambulation, while in seven out of 25 dogs, the study aimed to monitor the management of postoperative pain (Group A). Twelve dogs out of 30 recovered ambulation after being treated with surgery and EA, while in 8 out of 30, the study aimed to monitor the management of postoperative pain (Group B). Two out of 30 also received herbal medicine (Group B). The dogs had different grades of neurological dysfunction, and each is presented in Table 1.

The 1st, 2nd, and 6th studies were based on conventional therapy and were assigned to Groups C and D. One hundred forty-seven dogs were included, from which 66 were in Group C, 5 were in Group E, and a total of 76 dogs were treated in Group D. Twenty-seven out of 66 dogs from Group C had an improvement without neurological deficits; 17 out of 66 dogs from Group C walked without assistance, with only mild ataxia. Five dogs out of five in Group E had shorter days of ambulation recovery. Forty-nine out of 76 dogs from Group D recovered ambulance, while 20 out of 76 dogs from Group D walked without assistance, with only mild stumbling. The dogs had different grades of neurological dysfunction, and each is presented in Table 1.

## 4. Discussion

This review provides an outline of the relevant literature released over the last twenty years on dogs with intervertebral disc disease and therefore, EA’s effectivity in this species.

As a brief definition, WVM is more about the analysis and treatment of a disease process (infection, toxic or enzymatic disorders) using blood tests, imaging, drug, or even surgical procedures. WVM is also defined as conventional medicine. Conservative treatments use non-surgical options. Both conventional and conservative medicine is part of the Western world. Meanwhile, TCVM is about creating a balance Qi energy and is based on the body’s ability to regenerate [14].

In general, TCVM and WVM are just two different ways of seeing the world, each with its strengths and weaknesses. Their main purpose is to prevent and cure diseases. TCVM seems to offer an alternative in which Western medicine seems to have failed. The practice of TCVM in the Western world differs from its Chinese origins due to the continuous development of the system [14]. The most commonly used techniques in TCVM are described below.

Ac is a method that uses dry filiform needles introduced into certain acupoints. Recent studies show that acupoints occur where nerves enter tissues or where nerves branch [14].

EA uses an electric current passed through acupuncture needles that have already been inserted into the acupuncture points. EA is useful, especially in cases of neuralgia, but also in degenerative lesions of the nervous system. Some authors consider that a continuous, regular high frequency (80–120 Hz) mediates the release of endorphins and has an indication in the treatment of acute pain or muscle spasms, while a lower frequency (5–20 Hz) and alternative stimulation may be better for rehabilitating motor neurons in paresis and paralysis [14].

Another treatment regimen used in TCVM is herbal medicines. A good acupuncturist can predict his actions and prescribe the herbal formula for a certain clinical condition. Herb choice is made according to the plant temperature/energy, taste, energy direction, and channels introduced [15].

One of the most common treatments used for decompression of the spinal cord is hemilaminectomy. There are debates about the ideal moment for surgical intervention. When surgery was performed under 48 h after the onset of deep pain perception, 43% of dogs had a good prognostic and neurologic recovery. When surgery was performed after 48 h, the neurologic recovery was lower than 24%. When surgery cannot be an option after the loss of deep pain perception, EA might be considered an option for dogs with IVDD [9]. While spinal decompression surgery involves the physical removal of the cause, EA aims to stimulate nerve impulses to establish nerve connections, which can take days, weeks, or years, depending on the lesions or, of course, the patient [9].

Depending on the lesion location created by the intervertebral disc, whether it is single or multiple, besides the severity of the neurological dysfunction it produces, the foremost acceptable protocol is chosen [4,6].

In our review, only six studies classified the neurological deficiency of IVDD. The degree of neurological deficit was classified as grades from 1 to 5, in which grade 1 = only pain associated with medullary compression, grade 2 = conscious proprioceptive deficit and ambulatory para-paresis, grade 3 = non-ambulatory para-paresis with a deep pain, grade 4 = non-ambulatory paraplegia with a deep pain, and grade 5 = non-ambulatory paraplegia and no deep pain [6]. The rest of the studies mentioned only if the process was acute or chronic. Only one study reported the use of EA only, study 1, in which five dogs out of five had shorter days of recovery. We believe that there are insufficient data in the literature on quantifying EA’s effect in different grades of IVDD. However, it has been reported that the combination of corticotherapy with EA was significantly more effective than corticosteroid or acupuncture alone [4,16].

When evaluating this knowledge, it should of course be remembered that the absence of proof of effectiveness is not proof of the absence of effectiveness. As a result of most studies, there are no precise reasons for not adopting a scientific review approach in this area [16,17]. The known proof relates to various clinical presentations, very different acupuncture techniques, and outcome measures. Thus, at present, there is not enough evidence for any conditions [17,18].

Ac has been known to have good analgesic effects and could relieve back pain [10,11]. Ac could also activate axonal regrowth and the regeneration of destroyed axons within the funiculus. The quicker this regrowth occurs, the faster other axons might gain access to their original distal nerve fiber sheathes, due to less connective tissue at the lesion. One of Ac’s essential effects is anti-inflammatory because it can reduce local inflammation in the spinal cord and the side effects of histamine release. This significantly reduces spinal cord compression, the pain caused by possible fibrous tissue appearance, or even local hypoxia [7,8,19]. Salazar et al. [20] managed to provide strong support for EA’s use at specific immune points to stimulate mesenchymal stem cells (M.S.C.) and macrophage release into peripheral blood hypothalamic and the sympathetic nervous system. EA may facilitate tissue repair following injury by supplying high levels of circulating M.S.C. into the circulation, and it could be used to treat acute or chronic conditions associated with inflammation [21].

When acupuncture points are stimulated, nerve impulses will travel to the sensory nerves to enter the spinal cord. Most fibers are part of the pain pathway, although proprioceptive fibers are also part of acupuncture’s general activation. TCVM considers these channel connections, known as meridians. There seems to be a close connection between the meridians and the nerve pathways, especially those located in the extremities. A single mechanism cannot explain the effects of acupuncture. What starts as a local effect will spread to the nervous system and affect almost the entire body. Eventually, all the effects that started in the nervous system will cause changes in the endocrine and immune systems as well [14].

In combination with corticosteroids, the studies included in our review showed that it is a beneficial treatment option. The selected studies’ recovery rates were higher in Group D than in Group C, which could provide sufficient evidence to use it in clinical cases. Many experimental models of acute spinal cord concussion have demonstrated that corticosteroids have a neuroprotective effect when administered at the time of, or within minutes after, a spinal cord injury [22,23]. However, what ought to be emphasized is that solely short-term anti-inflammatory regimens of corticosteroids should be used. High doses of prednisone regimes should not be used after an anti-inflammatory treatment when a patient with compressive spinal disease acutely deteriorates because it favors gastrointestinal ulcers. Unfortunately, many patients with spinal injuries seen at referrals have already been treated with massive doses of steroid or nonsteroidal medicine, which force patients to face these negative effects. They may also influence the use and effects of corticosteroids [22,23].

Contrary to expectations, EA is becoming more widely used in the Western part of the globe (United States) compared to the Eastern part of the world (China and parts of Europe) [19]. EA is applied with different types of equipment that can offer a wide range of shapes and wavelengths, such as different frequencies and amplitudes. This type of therapy is currently widely used in human and veterinary acupuncture to treat various ailments, different types of pain, and even intraoperative pain. Compared to Ac, EA has the advantage of a shorter treatment time. The ability to stimulate can be adapted to the patient. Furthermore, EA produces much more stimulus (continuous flow and high amplitude) than can be provided manually [24]. Our reviewed articles suggested that the EA treatment option might be a promising technique when translated into clinical use.

Although it is a technique that could be considered, we must remember that rigorous training of the acupuncturist is required. Each acupoint has a precise position, and the treatment plan is chosen according to several parameters within which TCVM works. In the fourth study, the results were equivocal because the acupoints were chosen empirically, which may mean that the study was not conclusive [8].

When used together, Eastern (TCVM) and Western treatments offer better recovery and improvement of ambulation and perception of deep pain than medical treatment results alone. EA provides better analgesia than direct Ac by increasing spinal opioid release, and it increases blood supply to the spinal cord and nerve root [6,13].

Furthermore, EA application, combined with Chinese herbal medicine, should be an excellent therapeutic approach for postoperative care of patients with traumatic spinal cord injury [10,11,12]. In our review, we cannot forget about the effect of corticosteroids as well.

This is believed to be the first review of EA on dogs with spinal cord injuries. Therefore, this review’s limitations include the small number of studies meeting the inclusion criteria and the lack of intergroup comparisons. Another limitation that applies to reviews in general, is that all citations could not be followed. Therefore, some of them may not be complete. Although significant efforts have been made to collect all relevant data, some studies may not have been found. In our opinion, for future research, more attention should be paid to clinical cases than to the type of traditional Chinese treatment used because the last one is adapted to the patient.

## 5. Conclusions

The reported literature displays gaps due to either unexplored areas or insufficient findings. We should bear in mind that many studies have used a combination of EA and corticosteroids. Therefore, the anti-inflammatory effects of corticosteroids should be recognized as well.

Thus, more studies are warranted in this integrative medicine area because there is good potential for support. More specifically, appropriate protocols should be applied, depending on the patient’s condition and accurate diagnoses. Most importantly, from our perspective, these protocols should always be based on the consensus of experts.

## Figures and Tables

**Table 1 animals-11-00219-t001:** Neurological dysfunction recovery.

Study	Degree of Neurologic Dysfunction (Grade 1 to 5) (*n*. Dogs)	Acute (*n*. Dogs)	Chronic (*n*. Dogs)	Recovery of Ambulation (Outcome Measure) (*n*. Dogs)	Comments
1. Yang J, et al. 2003		Group C Gr 2 + 3:5		Group C Gr 2 + 3:5/5	
		Group D Gr 2 + 3:5		Group D Gr 2 + 3:5/5	Group D: recovery days were shorter than the other groups
		Group E Gr 2 + 3:5		Group E Gr 2 + 3:5/5	
2. Hayashi A, et al. 2007	Group C Gr 1 + 2:7 Gr 3 + 4:9 Gr 5:8			Group C Gr 1 + 2:7/7 Gr 3 + 4:6/9 Gr 5:1/8	
	Group D Gr 1 + 2:10 Gr 3 + 4:10 Gr 5:6			Group D Gr 1 + 2:10/10 Gr 3 + 4:10/10 Gr 5:3/6	Group D: Gr 5:2/6 dogs had spinal walking without deep pain
3. Hayashi A, et al. 2007			Group D Gr 3:1	Group D Gr 3:1/1	The dog also received Chinese herbal medicine
5. Joaquim J, et al. 2010			Group A Gr 4 + 5:10	Group A Gr 4 + 5:4/10	
			Group B Gr 4 + 5:11	Group B Gr 4 + 5:8/11	
			Group D Gr 4 + 5:19	Group D Gr 4 + 5:15/19	
6. Han H, et al.2010			Group C Gr 2 + 3:3737	Group C Gr 2 + 3:25/37	Group C: 17/25 dogs walked without assistance, only mild ataxia, 8/25 dogs without neurological deficits
			Group D Gr 2 + 3:4343	Group D Gr 2 + 3:39/43	Group D: 19/39 dogs recovered normal gait, 20/39 dogs walked without assistance, with only mild stumbling
7. Kim M, et al. 2011		Group B Gr 3 + 4:3		Group B Gr 3 + 4:2/3	Group B: all dogs also received Chinese herbal medicine
8. Kim S, et al. 2012	Group D Gr 5:1			Group D Gr 5:1/1	Group D: the dog has also received Chinese herbal medicine
9. Prado C, et al. 2019			Group A Gr 5:8	Group A Gr 5:4/8	Combining allogeneic stem cells derived from canine exfoliated deciduous teeth with EA was shown to be safe but brings only mild neurological and functional improvements.
			Group B Gr 5:8	Group B Gr 5:3/8	

Group A = surgery; Group B = surgery, and electroacupuncture (EA); Group C = corticotherapy; Group D = combination of corticotherapy and EA; Group E: EA. EA = electroacupuncture, Gr = grade, *n*. = number.

## Data Availability

Not applicable.

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
