# Peer review of "Current Aspects Regarding the Clinical Relevance of Electroacupuncture in Dogs with Spinal Cord Injury: A Literature Review"

_animals, 2021, doi:10.3390/ani11010219_

Round 1

Reviewer 1 Report

Dear Authors

Inclusion criteria: did the dogs underwent any surgical intervention? Did the studies include only non surgical patients? Were they experiencing acute or chronic issues?

Line 93: 10 treatment, I would like to have a more specific timetable of the treatments

I think you should report the different agopunture techniques used, you did mention as a critical point in the discussion but was not reported above

Author Response

Dear Reviewer,

I hope this e-mail finds you well. Here are the answers to your questions.

Point 1: Inclusion criteria: did the dogs underwent any surgical intervention? Did the studies include only non surgical patients? Were they experiencing acute or chronic issues?

Response 1: I have managed to change the classification in order to be easier to understand. (Line 72-75; Line 184-187)

 Point 2: Line 93: 10 treatment, I would like to have a more specific timetable of the treatments.

Response 2: The dog experienced recovery of proprioception and locomotion with only a slight deficiency in one of the thoracic limb after ten treatment sessions with EA and Chinese herbal medicine. The dog received 1 session of EA once a week for eight weeks; the last 2 sessions were made every two weeks; in total 10 sessions in 12 weeks (Line 110-113).

Point 3: I think you should report the different agopunture techniques used, you did mention as a critical point in the discussion but was not reported above

Response 3: .The most commonly used techniques in TCVM are described below.

AC is a method that uses dry filiform needles introduced into certain acupoints. Recent studies show that acupoints occurs where nerves enter tissues or where nerves branch.

EA is the use of electric current passed through acupuncture needles that have already been inserted into the acupuncture points. EA is useful especially in cases of neuralgia, but also degenerative lesions of the nervous system. Some authors consider that a continuous, regular high frequency (80-120 Hz) mediates the release of endorphins and has indication in the treatment of acute pain or muscle spasms, while a lower frequency (5-20 Hz) and alternative stimulation may be better for rehabilitating motor neurons in paresis and paralysis.

Another treatment regimen used in TCVM is herbal medicines. A good acupuncturist can predict his actions and prescribe the herbal formula for a certain clinical condition. The choice of herbs is made according to the temperature/energy of a plant, taste, as well as the direction of energy and Channels introduced. (Line 203-216).

Thank you for your time and considerations.

Reviewer 2 Report

Comments for authors

It is interesting to focus on the effectiveness of Electroacupuncture, but the group classification and evaluation methods in this review are ambiguous and difficult to understand.

Also, in the group classification, I would like you to clarify the relationship with surgical treatment. The same applies to the effect evaluation.

When concluding the effectiveness of Electroacupuncture, please indicate the effect of Electroacupuncture on pain control and the effect on motor function, respectively. About the mechanism that enables non-ambulatory dogs to walk without surgery.

[Current methods]

  1. Please set the group classification in this review, show the definition, and unify definitions throughout the review.
  2. What kind of treatments are included in Western medicine? Does this include surgery such as hemilaminectomy?
  3. Does Traditional Chinese treatment include electroacupuncture (Line 85-86)?
  4. Which group do “conventional medicine”, “western medicine”, “corticosteroid treatment”, “conservative treatment”, “medical treatment”, and “conventional analgesics” belong to?
  5. What is the deifference between “conventional medicine”, “western medicine” and “conservative treatment”?
  6. In group settings, pleas distinguish between combined surgical treatments. The same applies to the effect.

[Results]

  1. Please add a reference to each of study 1-9.
  2. Line 85-86: does traditional Chinese treatment include electroacupuncture?
  3. Line 93: what criteria is for marked improvement?
  4. Line 96-97: I don’t understand the classification criteria for group AB and C.
  5. Table1: Please show the effects of electroacupuncture on pain management and recovery of motor function by IVDD, divided into those without surgery and those with surgery.

[Discussion]

  1. In study 5, please evaluate the validity of the result that EA was more effective than surgery in the recovery of ambulation(Line 110-111).
  2. Please compare the effect of electroacupuncture with the effect of surgical treatment.
  3. From previous reports, please suggest treatment selection criteria for tetraplegia or grade 5 thoracolumbar disc herniation dogs.
  4. Prognosis and complications after surgical treatment are well reported, but electroacupuncture is poorly reported. Especially, when electroacupuncture is selected for tetraplegia or grade 5 thoracolumbar disc herniation dogs.
  5.  

Author Response

Dear Reviewer,

I hope this e-mail finds you well. Here are the answers to your questions.

Point 1: It is interesting to focus on the effectiveness of Electroacupuncture, but the group classification and evaluation methods in this review are ambiguous and difficult to understand.

Also, in the group classification, I would like you to clarify the relationship with surgical treatment. The same applies to the effect evaluation

Response 1: I have managed to change the classification in order to be easier to understand. (Line 72-75; Line 184-187)

Point 2: When concluding the effectiveness of Electroacupuncture, please indicate the effect of Electroacupuncture on pain control and the effect on motor function, respectively. About the mechanism that enables non-ambulatory dogs to walk without surgery.

Response 2: When acupuncture point are stimulated, nerve impulses will travel to the sensory nerves to enter the spinal cord. Most fibers are part of the pain pathway, although proprioceptive fibers are also part of the general activation of acupuncture. TCVM considers these channels connections known as meridians. There seems to be a close connection between the meridians and the nerve pathways, especially for those located in the extremities. The effects of acupuncture cannot be explained by a single mechanism. What starts as a local effect will spread to the nervous system and affect almost the entire body. Eventually, all the effects that started in the nervous system will cause changes in the endocrine and immune systems as well. (Line 247-256)

Point 3: Please set the group classification in this review, show the definition, and unify definitions throughout the review.

Response 3: I have managed to change the classification in order to be easier to understand. (Line 72-75; Line 184-187).

I also described the definition of each treatment used in the review. (Line 192-213)

 Point 4: What kind of treatments are included in Western medicine? Does this include surgery such as hemilaminectomy?

Response 4: Yes, Western medicine includes all the procedures we use in conventional medicine, such as medicines, surgical procedures, diagnostic methods, etc. (Line 192-195)

 Point 5: Does Traditional Chinese treatment include electroacupuncture (Line 85-86)?

Response 5: Yes, Traditional Chinese treatment include electroacupuncture as a treatment method (Line 200)

 Point 6: Which group do “conventional medicine”, “western medicine”, “corticosteroid treatment”, “conservative treatment”, “medical treatment”, and “conventional analgesics” belong to?

Response 6: I have managed to change the classification in order to be easier to understand. (Line 72-75; Line 184-187)

 Point 7: What is the deifference between “conventional medicine”, “western medicine” and “conservative treatment”?

Response 7: Western medicine is also defined as conventional medicine. Conservative treatment is using non-surgical options. Both terms, conventional and conservative medicine are part of Western world (Line 194-195)

 Point 8: In group settings, pleas distinguish between combined surgical treatments. The same applies to the effect.

Response 8: I have managed to change the classification in order to be easier to understand. (Line 72-75; Line 184-187)

Point 9: Please add a reference to each of study 1-9

Response 9: I have managed to add the references to each study.

 Point 10: Line 85-86: does traditional Chinese treatment include electroacupuncture?

Response 10: Yes, Traditional Chinese treatment include electroacupuncture as a treatment method (Line 200)

 Point 11: Line 93: what criteria is for marked improvement?                                     

Response 11: For the third study, I described the meaning for marked improvement: The dog experienced recovery of proprioception and locomotion with only a slight deficiency in one of the thoracic limb after ten treatment sessions with EA and Chinese herbal medicine. (Line 110-112)

 Point 12: Line 96-97: I don’t understand the classification criteria for group AB and C.

Response 12:  I have managed to change the classification in order to be easier to understand. (Line 72-75)

 Point 13: Table1: Please show the effects of electroacupuncture on pain management and recovery of motor function by IVDD, divided into those without surgery and those with surgery.

Response 13: I have managed to change the classification in order to be easier to understand. (Line 184-187)

 Point 14: In study 5, please evaluate the validity of the result that EA was more effective than surgery in the recovery of ambulation(Line 110-111)

Response 14: One of the most commonly treatment used for decompression of the spinal cord is hemilaminectomy. There are debates about the ideal moment for surgical intervention. When surgery was performed under 48 hours after onset of deep pain perception, 43% of dogs had good prognostic and neurologic recovery. When surgery was performed after 48 hours the neurologic recovery was lower than 24%. When surgery cannot be an option after loss of deep pain perception, EA might be considered an option for dogs with IVDD. (Line 214-219)

 Point 15: Please compare the effect of electroacupuncture with the effect of surgical treatment.

Response 15: The effects of acupuncture cannot be explained or compared by a single mechanism (Line 251-252). While spinal decompression surgery involves the physical removal of the cause, EA aims to stimulate nerve impulses to establish nerve connections, which can take days, weeks or years, depending on the lesions or, of course, the patients (Line 221-223).

 Point 16: From previous reports, please suggest treatment selection criteria for tetraplegia or grade 5 thoracolumbar disc herniation dogs.

Response 16: The degree of neurologic dysfunction was classified as grades from 1 to 5: grade 1 = only pain associated with IVDD; grade 2 = conscious proprioceptive deficit and ambulatory para-paresis, grade 3 = non-ambulatory para-paresis and deep pain perception, grade 4 = non-ambulatory paraplegia and deep pain perception; grade 5 = non-ambulatory paraplegia and no deep pain perception. (Line 226-230).

 Point 17: Prognosis and complications after surgical treatment are well reported, but electroacupuncture is poorly reported. Especially, when electroacupuncture is selected for tetraplegia or grade 5 thoracolumbar disc herniation dogs.

Response 17: We believe that there are insufficient data in the literature on quantifying the effect of EA in different grades of IVDD (Line 232-233).

Thank you for your time and considerations.

Round 2

Reviewer 2 Report

Comments to authors

Major comments

It became easier to understand than previous version after setting group. When describing the patient’s condition, please show the group name and grade of neurological dysfunction so that it is easier to interpret the results or effects.

Minor comments

Results

Line 109; “When both methods (EA and corticosteroid treatments)” is group D. I think it is better to indicate the group name.

Line 112ï¼›Please indicate the group name of the third study.

Line 121-122; Does “between the surgery group and the group that also received EA” indicate in group A and B?

Line 132-134; Is this indicate group D was more effective than group A?

Line 143; Does “All dogs” mean in group B?

Line 154-156; Please indicate the range of grades of neurological dysfunction.

Line 157-158; The dogs were in group E?

Line 160; Are “Five patients received only EA” in group E?

Line 161-162; Are “A total of 216 patients - - -” in group D?

Line 164; Are “The rest of the 131 dogs” in group C?

Line 138; Are “Three studies” in group E?

Line 172 “The 4th , 5th, 7th and 9th studies”; please indicate the range of grades of neurological dysfunction. Even if the grade was not written in the literature, you will be able to judge from the patient’s neurological information.

Line 180 “The 1st, 2nd and 6th studies”; please indicate the range of grades of neurological dysfunction.

Discussion

Line 140-148; The seventh study is all group B and has no comparison, how did you determine the effect of EA? I ask the same question about the 8th study (Line 149-153).

Line 236; You wrote faster, but what did you compare to?

Table 1; There is a lack of information of study 4. The place of the comment would be wrong in study 1. The grade of the dogs in study 9 did not match between the results description and the description in table 1.

Author Response

Dear Reviewer,

 Thank you for your time and suggestions, I hope that the manuscript is improved now.

Point 1: It became easier to understand than the previous version after setting the Group. When describing the patient's condition, please show the group name and grade of neurological dysfunction so that it is easier to interpret the results or effects.

Response 1:  The changes were made in the text accordingly.

Point 2: Line 109; "When both methods (EA and corticosteroid treatments)" is Group D. I think it is better to indicate the group name.

Response 2: The changes were made in the text accordingly.

Point 3: Line 112ï¼›Please indicate the group name of the third study.

Response 3: The group name was indicated: it is Group D.

Point 4: Line 121-122; Does “between the surgery group and the group that also received EA” indicate in groups A and B?

Response 4: Yes, I modified the text in order to be understandable.

Point 5: Line 132-134; Is this indicate group D was more effective than group A?

Response 5: Yes, I modified the text in order to be understandable.

Point 6: Line 143; Does “All dogs” mean in group B?

Response 6: Yes, they are in Group B. I modified the text.

 Point 7: Line 154-156; Please indicate the range of grades of neurological dysfunction.

Response 7: The whole paragraph was modified, and the grades of neurological dysfunction were indicated.

 Point 8: Line 157-158; The dogs were in group E?

Response 8: From a total of 226 dogs, 5 dogs had benefited of only electroacupuncture treatment and were included in Group E.

Point 9: Line 160; Are “Five patients received only EA” in group E?

Response 9: From a total of 226 dogs, 5 dogs had benefited of only electroacupuncture treatment and were included in Group E.

Point 10: Line 161-162; Are “A total of 216 patients - - -” in group D?

Response 10: A total of 216 patients (95.57 %) were treated with corticosteroids and were assigned to Group A, B, C, or D, depending on the treatment plan.

Point 11: Line 164; Are “The rest of the 131 dogs” in Group C?

Response 11: I have deleted the paragraph because does not bring scientific support for this review.

 Point 12: Line 138; Are “Three studies” in group E?

Response 12: The dogs were assigned to Group B and D. 

Point 13: Line 172 “The 4th , 5th, 7th and 9th studies"; please indicate the range of neurological dysfunction grades. Even if the grade were not written in the literature, you would be able to judge from the patient's neurological information.

Response 13: I have managed to indicate each study in which Group it is assigned. To be much easier to understand I indicated the range of grades of neurological dysfunction in Table 1.

 Point 14: Line 180 “The 1st, 2nd and 6th studies"; please indicate the range of neurological dysfunction grades.

 Response 14: I have managed to indicate each study in which Group it is assigned. To be much easier to understand I indicated the range of grades of neurological dysfunction in Table 1.

Point 15: Line 140-148; The seventh study is all group B and has no comparison, how did you determine the effect of EA? I ask the same question about the 8th study (Line 149-153).

Response 15: For the seventh study: It was considered a success when the dogs recovered from grade 5 to 2 of neurological dysfunction (Line 143).

For the eighth study: the dog had improved from grade 5 to grade 1 of neurological dysfunction with better mobility, proprioception, and spinal posture (Line 148-149)

Point 16: Line 236; You wrote faster, but what did you compare to?

Response 16: Only one study reported the use only of EA (Study 1), in which five dogs out of five had shorter days of recovery. I modified the word faster. (Line 231-232).

 Point 17: Table 1; There is a lack of information of study 4. The place of the comment would be wrong in study 1. The grade of the dogs in study 9 did not match between the results description and the description in table 1.

Response 17: Because study 4 was the only one that looked at pain management and not the recovery of ambulance as it was with the other articles, I have decided not to include it in Table 1 not to cause confusion between the articles. "Each study has a well-described treatment in Table 1, except for the 4th study, which was debated in the Discussion section.” (Line 91-93).

I modified the place of the comment in study 1. I modified the grade of the dogs for study 9.
